# Chronic Diseases and Employment: Which Interventions Support the Maintenance of Work and Return to Work among Workers with Chronic Illnesses? A Systematic Review

**DOI:** 10.3390/ijerph16101864

**Published:** 2019-05-27

**Authors:** Soja Nazarov, Ulf Manuwald, Matilde Leonardi, Fabiola Silvaggi, Jérôme Foucaud, Kristopher Lamore, Erika Guastafierro, Chiara Scaratti, Jaana Lindström, Ulrike Rothe

**Affiliations:** 1Health Sciences/Public Health, Faculty of Medicine Carl Gustav Carus, Technische Universität Dresden, Fetscherstraße 74, 01307 Dresden, Germany; soja.nazarov@tu-dresden.de (S.N.); ulf.manuwald@tu-dresden.de (U.M.); 2Neurologia, Salute Pubblica e Disabilità, FONDAZIONE IRCCS ISTITUTO NEUROLOGICO CARLO BESTA, 20133 Milan, Italy; matilde.leonardi@istituto-besta.it (M.L.); fabiola.silvaggi@istituto-besta.it (F.S.); erika.guastafierro@istituto-besta.it (E.G.); Chiara.Scaratti@istituto-besta.it (C.S.); 3Institut National du Cancer (INCa), 92100 Boulogne-Billancourt, France; jfoucaud@institutcancer.fr (J.F.); kristopher.lamore@gmail.com (K.L.); 4Health Education and Practices Laboratory (LEPS EA 3412), Paris 13, University—UFR SMBH—74 rue Marcel Cachin, 93017 Bobigny, France; 5Université de Paris, LPPS, 92100 Boulogne-Billancourt, France; 6Department of Public Health Solutions, Public Health Promotion Unit, National Institute for Health and Welfare, 00271 Helsinki, Finland; jaana.lindstrom@thl.fi

**Keywords:** return to work, chronic conditions, intervention, randomized controlled trial, systematic review

## Abstract

The increase of chronic diseases worldwide impact quality of life, cause economic and medical costs, and make it necessary to look for strategies and solutions that allow people with chronic diseases (PwCDs) to lead an active working life. As part of the CHRODIS Plus Joint European Action project, a systematic review was conducted to identify studies of interventions that support the maintenance of work and return to work (RTW) among workers with chronic illnesses. These interventions should target employees with the following conditions: diabetes, cardiovascular diseases, metabolic vascular syndrome, respiratory diseases, musculoskeletal disorders, mental disorders, and neurological disorders. An extensive search was performed in PubMed, EMBASE, and PsycINFO for English language studies. Included in this review were 15 randomized controlled trials (RCT) for adult employees (aged 18+). We found that workplace-oriented and multidisciplinary programs are the most supportive to RTW and reducing the absence due to illness. In addition, cognitive behavioral therapies achieve positive results on RTW and sick leave. Finally, coaching is effective for the self-management of chronic disease and significantly improved perceptions of working capacity and fatigue.

## 1. Introduction

More than one third of the European employable population has a chronic illness and two out of three people reaching retirement age have two chronic diseases [1]. About 15 million women and men between the ages of 30 and 70 die of non-communicable diseases (NCDs) every year worldwide [2]. The World Health Organization (WHO) defines chronic diseases as NCDs that have a long duration and are the result of a combination of genetic, physiological, environmental, and behavioral factors [2]. NCDs are the leading cause of mortality and morbidity in Europe and research suggests that complex diseases such as diabetes and depression will burden even more in the future [3]. The International Diabetes Organization (IDF) called the worldwide prevalence of diabetes a global epidemic [4]. With 425 million affected people (8.8% of the world’s population), the metabolic disease is becoming one of the world’s biggest health problems [5]. Further, depressive disorders were a leading cause of burden in the Global Burden of Disease (GBD) [6].

Europe pays a heavy price for chronic diseases (CDs): it is estimated that CDs cost EU economies 115 billion EUR or 0.8% of the Gross Domestic Product (GDP) each year. About 70% to 80% of health budgets in the EU are spent on the treatment of CDs [7].

CDs are not only widespread in the workforce but often lead to relevant impairment of work activities and work participation among affected workers. For example, CDs are the second most common cause of work incapacity, resulting in twice as many absences as other illnesses, and are the most common causes of disability pensions [8].

Despite problems in the workplace, most workers with CDs can lead a productive life; from an economic and social point of view, the participation of these people in the labor market will be imperative in the future. Various factors related to the disease itself or irrespective of diagnosis influence the process of work participation. Previous research indicates that work participation is influenced by both personal factors (e.g., age, education, gender, self-assessment of the ability to work) and work-related factors (e.g., heavy manual work, perceived control over work situation) [9].

“Return to work” (RTW) is the internationally accepted term for all activities that enable and facilitate returning to work after an illness [10]. These activities can be people-oriented or workplace-oriented intervention programs, rehabilitation programs, and training tools, including for example, cognitive behavioral therapy, increasing activity, workplace adaption, etc.

As part of the CHRODIS Plus European Joint Action project, a systematic review was performed to identify and analyze which interventions support the maintenance of work and RTW among workers with chronic illnesses. These interventions should target workers with CDs in general and more specifically workers with the following diseases: diabetes, cardiovascular diseases (CVD), metabolic vascular syndrome (MVS), respiratory diseases, musculoskeletal disorders, mental disorders, and neurological disorders. Cancer was investigated by the French National Cancer Institute (INCa) and is published elsewhere [11].

These diseases were included because of their prevalence, burden on health services and labor organizations, and their potential to limit working ability or to cause disability, absenteeism, and early retirement. Evidence from Germany demonstrates that the largest share of disability and early retirement are due to musculoskeletal diseases and mental and behavioral disorders, and that these diseases have a significant impact on ability to work, absenteeism, and medical and economic costs. Regarding early retirement, CVD are the third most common cause [12]. There is also growing evidence to suggest that those with physical illnesses, such as heart disease, back pain, or cancer, may also show mild to moderate symptoms of depression. This can also affect their work and well-being upon returning to work after long term sickness absence [13].

Although the cost of medical treatment caused by diabetes are very high, the costs of absenteeism and reduced ability to work are rather low [14]. However, according to IDF, diabetes and MVS play a major role as risk factors for CVD [5].

Among respiratory diseases, chronic obstructive pulmonary disease (COPD) and pulmonary emphysema are associated with the greatest loss of working days. COPD is currently the fourth leading cause of death worldwide and the Global Burden of Disease study predicts that COPD will be the third leading cause of death in 2020 [15]. COPD is also associated with a 2.5 times higher risk for CVD [16]. Finally, neurological disorders, like stroke, are often combined with CVD.

## 2. Materials and Methods

A review protocol was created a priori in which the search strategy, article selection, data extraction, and synthesis have been described. The Preferred Reporting Items for Systematic Reviews (PRISMA statement) has been used as a formal systematic review guideline [17].

### 2.1. Search Strategy

Since this review was done as part of the CHRODIS Plus European Joint Action project, all members of this project agreed on the methodology and decided to search the three main medical databases: PubMed, EMBASE, and PsycINFO. An extensive search was performed in PubMed until 31 March 2018, EMBASE until 1 April 2018, and PsycINFO until 2 April 2018. We searched for all English language studies, using terms or synonyms for chronic diseases and RTW (see Box 1). Research was done using free text and keyword searches to increase the sensitivity of the search string and using filters like “humans” for higher specificity. The search terms were linked using Boolean operators “AND” and “OR”. The strategy was formulated in PubMed (MEDLINE) and was adapted for use in the other databases.

Box 1Search terms for PubMed.(Chronic disease [MeSH] OR chronic diseas* OR chronic condition* OR multiple chronic conditions [MeSH] OR multiple chronic diseases OR multimorbidity [MeSH] OR multimorbid*) AND (return-to-work OR return to work/organization and administration [MeSH] OR return to work/statistics and numerical data [MeSH] OR re-integrating OR back to work OR employment [MeSH] OR employment sector OR sick leave [MeSH] OR absenteeism [MeSH] OR occupational medicine [MeSH] OR occupational health [MeSH] OR occupational health services [MeSH] OR “disability management” OR “disability prevention” OR employer*) AND (rehabilitation [MeSH] OR rehabilitation program OR training program* OR training tool* OR training OR occupational rehabilitation OR occupational intervention OR workplace intervention OR occupational therapy OR stress management OR work ability) AND (randomized controlled trial [MeSH] OR randomized controlled trial OR controlled clinical trial OR controlled clinical trial [Publication Type] OR evaluation study OR evaluate* OR effects OR effectiveness OR efficiency OR process OR outcome)

### 2.2. Inclusion and Exclusion Criteria

Randomized controlled trials (RCTs) and controlled clinical trials (CCTs) were included. Studies were selected if they described factors related to RTW of employed adults (aged 18+) with CDs in general or one of the following CDs: diabetes, CVD, MVS, respiratory disease, mental diseases, musculoskeletal disorders, and neurological disorders. The search was carried out without temporal and geographical limitations. Excluded were meta-analysis, reviews, cohort studies, crossover studies, case-control studies, cross-sectional studies, and programs that were not evaluated or tested with a comparison group.

### 2.3. Selection of Studies

All search results were managed with EndNote X7.1 (Clarivate Analytics, Philadelphia, PA, USA). After the removal of duplicates, title screening and abstract screening were performed independently by two researchers. The screening was done using the predefined inclusion and exclusion criteria. When the title and abstract did not fulfill one or more selection criteria, the record was excluded. In the second round of screening, full text articles were read and selected independently by two reviewers based on the predefined criteria.

### 2.4. Quality Assessment

Within the framework of the CHRODIS Plus project, two systematic reviews were conducted: (1) this review on RTW and chronic diseases in general and (2) a previously published review on RTW and cancer [11]. The review of RTW among cancer patients did not find enough RCTs, so the colleagues decided to also include other study designs. Otherwise, both reviews should, as far as possible, use a common methodology. For this reason, it was decided that both systematic reviews should assess methodological quality using the Critical Appraisal Skills Program (CASP) 2017 checklists, because these are available for various study designs.

The CASP methodological quality assessment for RCTs comprised the three sections: “Are the results of the trial valid?” (Section A); “What are the results?” (Section B); and “Will the results help locally?” (Section C) [18]. The CASP checklist results in a total score ranging from 1 to 11 (best) for each study. Based on the achieved score, we decided to rate each study with 1 to 5 “yes” points as a study of low quality (i.e., participants were recruited in an unacceptable manner, unclear handling of follow-up, and poorly defined end results), each study with 6 to 8 “yes” points as a study of good quality (i.e., participants were recruited in an acceptable manner, clear handling of follow-up and poorly defined end results), and each study with 9 to 11 “yes” points as a study of very good quality (i.e., the participants were recruited in an acceptable manner, clear handling of follow-up and clearly defined end results).

Based on our clearly defined question, study population, allocation of participants, data analysis, intervention effect, and handling of failure rates, the studies had with a 92.4% agreement among raters a good to very good quality. Six studies were rated with scores of 11 and were found to be of very good quality [19,20,21,22,23,24]. The remaining nine studies [1,25,26,27,28,29,30,31,32] were of good quality; no study was rated as a study of poor quality. The Christensen et al. study [31] scored the fewest points because of the lack of clear statements about randomization, blinding of participants in the allocation, and intervention. Table 1 shows the methodological quality of the studies included in our review.

### 2.5. Data Extraction and Synthesis

Data were extracted by one reviewer and checked by the other reviewers. The extracted study characteristics (author, year, country, study design), patient characteristics (CDs diagnosis, number of included participants, age of participants, gender of participants, education and employment status), primary and secondary outcomes, and follow-ups are reported in Appendix A. Extracted intervention characteristics (type of intervention, aim of intervention, content of intervention, number, and discipline of trainer or counsellors) [33] are reported in Appendix A.

## 3. Results

A total of 2560 records were yielded through the search strategy: 1875 from PubMed, 582 from EMBASE, and 103 from PsycINFO. After excluding duplicate records, 2264 records remained. Performing a title and abstract screening excluded 2189 records and identified 75 records for a full-text analysis. Based on the full-text selection, 13 studies met the inclusion criteria and were included in this review. Checking the reference list of the 13 studies yielded two additional records. The research work flow (15 studies) and study selection are presented in Figure 1.

### 3.1. Study, Participants, and Interventions Characteristics

The 15 RCTs included in this report were published between 2000 and 2017, mostly in Europe (Table 2).

Among the participants, 46% of the employed persons had musculoskeletal diseases followed by 41% with mental illnesses. Participants with different CDs were described in two studies [1,26]. CVD and neurological disease (stroke) were only reported in one South African study [23].

The participants in the included studies had different working status: 260 (13%) were in paid work, 779 (38%) were absent due to illness, and 172 (8%) of the participants were unemployed. Work status was not reported in two studies [20,24].

Different interventions for chronically ill workers were identified. These varied both in terms of the objectives and use of the intervention programs, as well as in relation to the comparison strategies and diseases.

Christensen et al. [31] compared three different intervention programs; Bergström et al. [27] compared three different intervention programs with a control group; Dalgaard et al. [29] compared an intervention group with two control groups; and Nieuwenhuijsen et al. [22] used both a control group and a placebo group. Two different occupational rehabilitation programs were compared in the study by Friedrich et al. [20]. In comparison, the remaining 10 studies use the control groups with treatment as usual [1,19,21,23,24,25,26,28,30,32].

Participation rates at the start of the studies were noted. The follow-up varied between 6 weeks and 10 years.

Appendix A shows the characteristics of the included studies. Table 2 summarizes the characteristics of the included studies.

### 3.2. Effects of Interventions

Our research question defined the maintenance of work and RTW of chronically ill people as the primary outcome. This can be measured as RTW rate, RTW time, RTW percent, duration of sickness absence, sick leave in days, and working ability. Statistical methods for evaluating the results vary and accordingly the effect sizes (Table 3).

We did not have predefined secondary outcomes. However, job satisfaction, quality of life, improvement in functional status, physical activity, and fatigue and pain intensity may have a positive or negative effect on the primary results. Therefore, these were included in our analysis. de Buck et al. [19] examined work loss (total incapacity or unemployment) and Detaille et al. [1] examined self-efficacy at work and attitudes toward self-management at work as their primary outcomes. There were no secondary findings in six studies [21,26,27,29,31,33].

#### 3.2.1. Workplace-Oriented Intervention Programs

Four studies [23,25,28,30] examined workplace-oriented intervention programs. Ntsiea et al. [23] found that a workplace intervention consisting of work ability assessments and working visits facilitated the resumption of work for stroke survivors in the Gauteng province of South Africa. After a follow-up of six months, 60% (*n* = 24) of stroke survivors returned to work in the intervention group, compared with 20% (*n* = 8) in the control group (*p* < 0.001). The odds ratio for RTW for stroke survivors in the intervention group was 5.2.

Varekamp et al. [25] found that empowerment training can increase self-efficacy and help reduce fatigue problems, which in the long run could lead to more job retention. The training program focused on solving work-related issues. A step-by-step approach was adopted: first, work-related issues were investigated and clarified; second, communication at work was addressed; and third, solutions were developed and realized. It is reported here that self-efficacy increased at 24 months and fatigue decreased more significantly in the experimental group than in the control group (10 versus 4 points (*p* = 0.000) and 19 versus 8 points (*p* = 0.032)).

Lambeek et al. [28] compared an integrated care program that combines patient and workplace interventions for patients with chronic low back pain with standard care, and found significant results for the effectiveness of the intervention. The median duration to sustainable RTW was 88 days in the integrated care group compared to 208 days in the usual care group (*p* = 0.003). The integrated care effectively increased the RTW rate (Hazard ratio (HR) 1.9, 95% CI 1.2 to 2.8). Kim Wong et al. [30] found that a supported employment program based on the individual placement and support model for the unemployed with long-term mental illnesses was very positive in terms of work (70% versus 29%; odds ratio (OR) 5.63, 95% CI 2.28 to 13.84).

#### 3.2.2. Cognitive Behavioral Therapy Interventions

Two studies [27,29] examined cognitive-behavioral intervention programs. Dalgaard et al. [29] investigated the efficacy of work-oriented cognitive-behavioral therapy (CBT) combined with an optional workplace intervention for people with work-related stress symptoms. The researchers found that the median number of weeks to permanent RTW was 15, 19, and 32 for the intervention, control A, and control B groups, respectively. In the fully adjusted Cox regression model, the intervention group showed significantly faster sustained RTW at 44 weeks (HR 1.57, 95% CI 1.01 to 2.44) relative to control group A. In the intervention group there was a tendency for faster RTW compared to control A. The intervention group returned to work about 4 weeks earlier than control group A.

The effect of physical therapy (PT), cognitive behavioral therapy (CBT), behavioral medicine rehabilitation (BM), and a “treatment-as-usual” on absenteeism due to chronic neck pain (NP) and/or back pain (LBP) was studied over a 10-year period by Bergström et al. [27]. Three different groups of patients were derived empirically from the Swedish version of the Multidimensional Pain Inventory (MPI-S): interpersonally distressed (ID) patients who are characterized by low social support, dysfunctional (DYS) patients with high pain severity, disability, and affective distress, as well as adaptive copers (AC) who report a more successful adaptation to chronic pain. The three groups of patients underwent four different interventions. In terms of long-term follow-up of sick leave, BM was most advantageous for DYS and AC patients. In contrast, CBT and PT interventions did not benefit any patient population.

#### 3.2.3. Self-Management Programs

An adaptation of the Stanford University’s chronic disease self-management program (CDSMP) was studied by Detaille et al. [1]. The original CDSMP focuses on personal lifestyle factors and disease-related factors, such as coping with disease symptoms. The adapted program included work-related factors, such as communicating with colleagues and supervisors, acquiring resources at work, and dealing with disease symptoms at work. The researchers found that the attitude towards self-management at work (fun) improved after eight months for the intervention group (p = 0.030). No other outcome variable was significantly different.

#### 3.2.4. Vocational Rehabilitation (VR) Programs

De Buck et al. [19] evaluated the effectiveness of a multidisciplinary program for reducing the risk of loss of work among employees with chronic rheumatic diseases. At 24 months, no difference was found for the risk of job loss between groups.

#### 3.2.5. Coaching Interventions

There have been two studies of coaching intervention programs [22,26]. McGonagle et al. [26] tested a 12-week, six-session telephone coaching intervention designed to help employees meet challenges and reduce stress. Using stress and resource theories, it was believed that coaching would boost workers’ internal resources and lead to better perceptions of work ability, exhaustion and exclusion burnout, self-efficacy at work, core self-esteem, resilience, mental resources, and job satisfaction, and that these positive effects would remain stable 12 weeks after coaching. Compared with the control group, the coaching group showed significantly improved perceptions of work ability, fatigue, self-esteem, and resilience. However, no significant improvements were found in workplace self-efficacy, burnout, or job satisfaction.

In the study by Nieuwenhuijsen et al. [22], a new treatment platform with light therapy plus Pulsed Electro Magnetic Fields (PEMF) was used in combination with coaching for workers with work-related chronic stress symptoms compared to coaching alone. All groups improved significantly over time in terms of RTW.

#### 3.2.6. Comparative Intervention Strategies

Comparative intervention strategies were evaluated in three studies: Bendix et al., Christensen et al., and Friedrich et al. [20,31,32].

The effects of a comprehensive program of functional recovery, intensive physical training, ergonomic training, and behavioral support (39 hours a week for 3 weeks) for patients with chronic LBP compared to outpatient intensive physical training (1.5 hours three times a week for 8 weeks) were examined by Bendix et al. [32]. At the one-year follow-up, the overall score favored functional recovery. Elsewhere, there were no significant differences in work ability, sick leave, health care contact, back pain, leg pain, or self-reported daily activities.

The effects of three different rehabilitation strategies (video, back café, or Training) were observed by Christensen et al. [31]. Video group participants watched a video of exercises for training and were provided instructions regarding their use only once. The back café group was provided the same video program, but as a supplement met with other fusion-operated patients at a back café three times over an eight-week period. The training group was provided physical therapy training twice weekly for eight weeks. After two years of follow-up, more surgery patients worked again in the back café group compared to the other two groups (*p* < 0.04).

The study by Friedrich et al. [20] compared a training program that combined exercise and a motivation program on the disability level of patients with chronic and recurrent LBP with an exercise program alone. The exercise program consisted of submaximal, gradually increased exercises. The treatment was designed to improve spine mobility, muscle length, strength, endurance, and coordination of the trunk and lower limbs by restoring normal function. The motivational program consisted of the following interventions: (1) comprehensive counseling and information strategies, (2) reinforcement techniques, (3) oral agreements between the patient and the therapist were confirmed in writing in the form of a “treatment contract”, (4) patients were asked to complete the treatment contract, and (5) finally, patients were more involved in their care by reporting all the exercises they had done in an exercise diary. Five years after the supervised combined exercise and motivation program, patients had significant improvements in disability, pain intensity, and work ability. A significant positive long-term effect on the five-year workability re-evaluation was observed only in the motivational group.

#### 3.2.7. Interventions That Prevent or Slow Down Chronicity

Two studies described interventions intended to prevent or slow chronicity [21,24]. Bakker et al. [23] described the “minimum intervention for stress-related mental disorders with sick leave” (MISS) for employees with stress-related mental disorders, which was intended to reduce the duration of illness. MISS had no overall effect on the duration of the sick leave (HR 1.06, 95% CI 0.87 to 1.29) or on the severity of self-reported symptoms. No evidence was found that the MISS is more effective than usual care on in the study sample of distressed patients.

Linton et al. [21] compared a CBT intervention for people with chronic low back pain with providing one of two different types of information: (1) Participants received a previously evaluated pamphlet to read concerning back pain. The pamphlet provided straightforward advice about the best way to cope with back pain by remaining active and thinking positively (pamphlet group); (2) The information package group received a packet of information once a week for six weeks. Each package contained advice and illustrations showing how the patient might cope with spinal pain or prevent it by lifting properly and maintaining good posture (information package group).

All three comparison groups reported successes. The risk of long-term sick leave was reduced only for the CBT intervention group; the relative risk was nine times lower compared to the information package groups (relative risk, 9.3). In addition, the CBT group showed a significant decrease in the use of physicians and physiotherapy compared to the two groups that received information (where such use was increased). All three groups showed improvements for the variables pain, anxiety prevention, and cognition.

### 3.3. Secondary Outcome

Fatigue is a commonly referred complaint to chronically ill patients, especially when they have a job [34]. The multidisciplinary job retention vocational rehabilitation (VR) program has shown significantly greater improvement in fatigue and emotional status in patients receiving VR over a 24-month period compared to usual care (all *p*-values <0.05). Similarly, a job-preservation program achieved an increase in self-efficacy and significant decrease in fatigue in an intervention group after 24 months, which in the long run could lead to more job-preservation [19].

An integrated care program that combines patient- and workplace-based interventions for patients with chronic low back pain significantly improved the functional status of patients in the integrated care group at 12 months compared to patients in the usual care group (*p* = 0.01) [28]. The improvement in pain between the groups did not differ significantly.

Significant improvements in disability and pain intensity were achieved through combined movement and motivation programs at the disability level of patients with chronic and recurrent LBP [20].

Apart from increasing the chances of RTW, cognitive ability can also increase quality and activities of daily life, as shown by Ntsiea et al. [23]. Moreover, for every increase in the unit of daily life activities or cognitive score, the chances of resuming work after a stroke increased by 1.7 and 1.3, respectively.

In four studies [26,27,31,32], no significant secondary outcomes were found in terms of quality of life, fatigue, and improvement in functional status.

## 4. Discussion

The aim of this study carried out in the framework of the CHRODIS Plus European Joint Action project was to give a summary of the available interventions that aim to enhance or facilitate work reintegration among people with chronic diseases. We included 15 RCTs in this review that were published between 2000 and 2017 in English. Seven studies included workers with musculoskeletal disorders, five included workers with mental health disorders, two included workers with various chronic illnesses, and one included people affected by CVD and stroke. No studies were found for MVS. Most of the studies were published in the Netherlands and in Nordic countries of Europe, namely Denmark and Sweden.

The RCTs included in our review reported on interventions aimed both at improving personal skills and improving labor market participation.

We found that workplace-based interventions can lead to positive changes in employment status, work ability, RTW, and sick leave rates for people with various chronic conditions [23,25,28,30]. In addition, an increase in functional status and quality of life, as well as a reduction in pain intensity and fatigue were observed following the interventions. This suggests that a generic approach may be considered to improve the participation at work of people with various chronic conditions. This statement is also confirmed by an overview by Vooijs et al. [9]. Most of the reviews in the overview by Voojs et. al. [9] reported positive outcomes for work-related interventions. The reduction in sick leave rates among workers with musculoskeletal disorders and the impact on health through workplace measures were also demonstrated in a systematic review by Van Vilsteren et al. [35]. Although the applied interventions included participants with various chronic diseases, it cannot be excluded that patients with certain chronic conditions would benefit less or more from certain intervention components.

Multidisciplinary interventions are promising strategies that can meet the complexity of the needs of people with chronic conditions [35]. They are characterized by teams that include several professionals from different sectors (e.g., as physicians, occupational physicians, psychotherapists, occupational therapists) [36]. An integrated care program by Lambeek et. al. [28] provided significant results for RTW rate and for improvement in functional status and a multidisciplinary program provided a significant effect on reducing fatigue and improving mental health [18]. Similar results were reported by researchers in the recently published systematic review by Sabariego et al. [36]. Here, four of the seven studies on multidisciplinary interventions reported positive outcomes. Additionally, in a systematic review by Hoefsmit et al. [37], it was found that the multidisciplinary interventions are effective in supporting RTW in multiple audiences (e.g., back pain and adjustment disorders). The factors that positively or negatively affect the multidisciplinary intervention need to be further researched.

In coping with work-related stress, stress management interventions have proven to be as effective as cognitive-behavioral interventions. These consistently produced greater effects than other types of interventions [38]. In our review, we found that an individual’s work-oriented CBT with intervention on workplace for employees with adjustment disorder or reactions to severe stress can lead to significant improvement of permanent RTW and was the most beneficial in terms of long-term follow-up on sick leave.

The return to work for chronically ill patients can be improved by developing their ability to purposefully influence their own behavior. Self-management is an approach increasingly used in chronic disease care to improve self-efficacy and well-being. Detaille et al. [1] found self-management programs could not significantly improve self-efficacy at work, job satisfaction, and the intention to change jobs. However, the result might be explained by many factors: small number of participants, low level of education, and different diagnoses.

Coaching through education, behavioral change, or psychosocial approaches seems to play a significant role in improving the self-management of chronic diseases [39]. The McGonagle et al. [26] study tested a 12-week telephone coaching intervention designed to help employees meet challenges and reduce stress. Using stress and resource theories it was assumed that coaching promoted the internal resources of the workers and would lead to a better perception of the ability to work, burnout exhaustion and exclusion, self-efficacy at work, core self-evaluations, resilience, mental resources, and job satisfaction and that these positive effects would remain stable 12 weeks after coaching. Here, it was found that the coaching group showed significantly improved perceptions of working capacity, fatigue, self-assessments, and capacity compared to the control group. However, no significant improvements in workplace self-efficacy, burnout, or job satisfaction were found.

Nieuwenhuijsen et al. [22] investigated the effect of combining mental coaching with light therapy plus pulsed electromagnetic field therapy versus mental coaching alone. All groups improved significantly over time in terms of return to work. This positive result for both groups may be attributed to mental coaching alone. Moreover, combined exercise and motivation programs can achieve good long-term effect on pain intensity and working capacity of patients with chronic and recurrent back pain.

Although research on coaching for self-management of chronic illness is limited, existing studies suggest that coaching is effective in the self-management of chronic disease [39]. Combined exercise and motivation programs can improve long-term effects on disability, pain intensity, and work capacity of patients with chronic and recurrent back pain [20].

Some limitations should be mentioned. Although the choice to include only RCTs guaranteed a strong study design, it did not allow us to consider many observational studies that also focused on the effectiveness of interventions, with the consequent loss of some results.

Another limitation was that we were looking for literature in English, although some publications of interest may also be published in the languages of the countries where they were performed.

Finally, due to the scope of this work, we focused on employment outcomes; other important results that may have had an impact on intervention results, such as health and quality of life, were omitted.

Despite these limitations, the study presents the following strengths: (1) we performed a methodological quality assessment of the study and (2) we conducted the search of relevant studies without any time and geographical restrictions, which permitted us to consider a broader range of interventions. The results from our review are very heterogeneous and needs further research. Nevertheless, our results provide a part for the development of a training tool of RTW in the CHRODIS Plus project.

## 5. Conclusions

Various interventions targeting populations with very different CDs were identified. These interventions focused mainly on changes in work and most of them are effective in improving work participation. This indicates that work-oriented interventions should be considered as a general approach to improve work participation of employees with various CDs.

Coaching has great potential for improving self-management in chronic conditions. The literature on this, as well as in our systematic review, is limited. Future research is needed to look at the mechanisms that may make coaching successful in reducing the burden of chronic disease.

In dealing with work-related stress, stress management interventions such as cognitive-behavioral interventions have proven effective. Individualized work-oriented CBT with an optional workplace intervention for those with adjustment disorder or reactions to severe stress can significantly improve long-term RTW. In addition, CBT can reduce the risks of long-term sick leave.

The best strategy for RTW or for improving functional status, mental health and fatigue are multidisciplinary interventions involving various health and work professionals. Therefore, further multidisciplinary intervention programs should be researched and developed.

## Figures and Tables

**Figure 1 ijerph-16-01864-f001:**
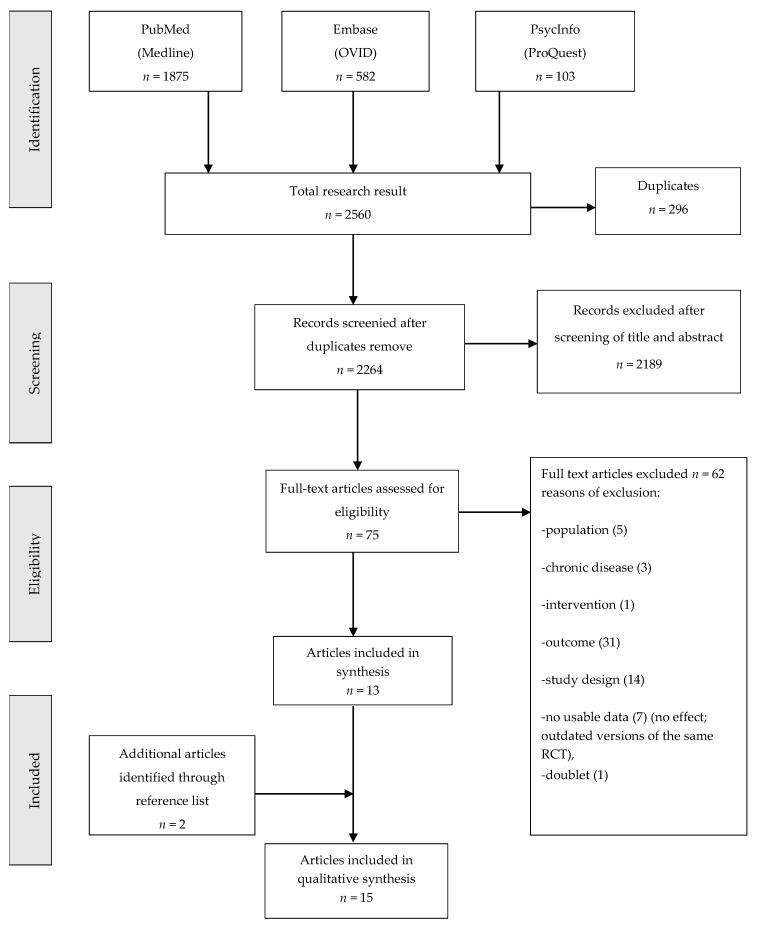
Flow diagram of study selection according to the Preferred Reporting Items for Systematic Reviews (PRISMA).

**Table 1 ijerph-16-01864-t001:** Critical Appraisal Skills Program (CASP): Authors agree on methodological quality items for each included study.

Author	CASP Questions (Q)											Judgement		
	Section A ^1^						Section B ^2^		Section C ^3^			Score ^4^		
	Q1	Q2	Q3	Q4	Q5	Q6	Q7	Q8	Q9	Q10	Q11	∑ yes	∑ no	∑ can’t tell
Bakker et al. [24]	+	+	+	+	+	+	+	+	+	+	+	11		
Bendix et al. [32]	+	+	+	−	+	+	+	+	+	+	−	9	2	
Bergström et al. [27]	+	+	+	+	+	+	+	+	+	+ −	+	10		1
Christensen et al. [31]	+	+ −	+	+ −	+ −	+	+	+ −	+	+	+ −	6		5
Daalgard et al. [29]	+	+	+	+ −	+	+	+	+	+	+	+	10		1
De Buck et al. [19]	+	+	+	+	+	+	+	+	+	+	+	11		
Detaille et al. [1]	+	+	+	+	+	+	+	+	+	+ −	+ −	9		2
Friedrich et al. [20]	+	+	+	+	+	+	+	+	+	+	+	11		
Lambeek et al. [28]	+	+	+	−	+	+	+	+	+	+	+	10	1	
Linton et al. [21]	+	+	+	+	+	+	+	+	+	+	+	11		
McGonagle et al. [26]	+	+	−	+ −	+	+	+	+	+	+	+	9	1	1
Nieuwenhuijsen et al. [22]	+	+	+	+	+	+	+	+	+	+	+	11		
Ntsiea et al. [23]	+	+	+	+	+	+	+	+	+	+	+	11		
Varekamp et al. [25]	+	+	+	−	+ −	+	+	+	+	+	+ −	8	1	2
Kin Wong et al. [30]	+	+	+	−	+	+	+	+	+ −	+	+ −	8	1	2

Legend: “+”―yes: “- “―no; “+ -“―can’t tell. ^1^ Are the results of the study valid? ^2^ What are the results? ^3^ Will the results help locally? ^4^ Methodological quality “yes” scores: 6–8 (good), 9–11 (very good); CASP Questions legend: Q1 = “Did the trial address a clearly focused issue?”; Q2 = “Was the assignment of patients to treatments randomized?”; Q3 = “Were all of the patients who entered the trial properly accounted for at its conclusion?”; Q4 = “Were patients, health workers and study personnel ‘blind’ to treatment?”; Q5 = “Were the groups similar at the start of the trial?”; Q6 = “Aside from the experimental intervention, were the groups treated equally?”; Q7 = “How large was the treatment effect?”; Q8 = “How precise was the estimate of the treatment effect?”; Q9 = “Can he results be applied to the local population, or in your context? ”; Q10 = “Were all clinically important outcomes considered?”; Q11 = “Are the benefits worth the harms and costs?

**Table 2 ijerph-16-01864-t002:** Summary of the characteristics of the included studies.

	Number of Studies	Number of Participants
**Country of publication**		
Netherlands	6	
Denmark	3	
Sweden	2	
Austria	1	
USA	1	
South Africa	1	
China	1	
**Intervention adapted to one of chronic disease**		
Diabetes	0	0
Cardiovascular diseases and neurological disease (stroke)	1	80 (4%)
Metabolic vascular syndrome	0	0
Respiratory disease	0	0
Mental diseases	5	851 (41%)
Musculoskeletal disorders	7	956 (46%)
Different chronic disease	2	181 (9%)
**Interventions**		
Workplace oriented intervention programs	4	
Cognitive behavioral therapy interventions	2	
Self-management programs	1	
Vocational rehabilitation programs	1	
Coaching interventions	2	
Comparative intervention strategies	3	
Interventions that prevent or slow down chronicity	2	
**Participants (N included in the intervention)**		
Total		2068
female		1319 (64%)
male		749 (36%)

**Table 3 ijerph-16-01864-t003:** Summary of the primary outcome. RTW: return to work; CBT: cognitive-behavioral therapy; PEMF: Pulsed Electro Magnetic Fields.

Outcome/Diseases	Intervention	Study *n*	Participants *n*	Statistical method	Effect size
1. RTW (lasting RTW, RTW rate, RTW percentage, employment rate, work resumed)		5			
1.1. Adjustment disorder	Work focused CBT	1	163	Hazard Ratio (95% CI)	1.7 (1.01 to 2.44)
1.2. Work-related chronic stress	PEMF	1	84	Mean difference *p*-value	0.92
1.3. Stroke	Work place intervention	1	80	Odds ratio (95% CI)	5.2 (1.8 to 15.0)
1.4. Long-term mental illness	Supported employment program	1	92	Odds ratio (95% CI)	5.63 (2.28 to 13.84)
1.5. Isthmic spondylolisthesis	Three different rehabilitation programs	1	90	Mean difference *p*-value	0.04
2. Work ability		4			
2.1. Chronic low back pain	2.1.1. Functional restoration program	2	99	Mean difference *p*-value	0.64
	2.1.2 Vocational rehabilitation		56	Mean difference *p*-value	0.005
2.2. Chronic rheumatic disease	Job-retention vocational rehabilitation program	1	140	Mean difference *p*-value	0.13
2.3. Different diseases	Phone-based coaching	1	59	Pre-mean (SD)	3.39 (0.75)
				Post-mean (SD)	3.82 (0.39)
3. Sick leave until full RTW, long-term sick leave, sickness absence		4			
3.1. Stress related mental disease		1	433	Hazard Ratio (95% CI)	1.06 (0.87 to 1.29)
3.2. Chronic low back pain	3.2.1. Different intervention (behavioral oriented physiotherapy, cognitive behavioral therapy, behavioral medicine rehabilitation) using a psychosocial subgroup	2			
			194	mean difference, (95% CI)	BM: −16.08, (−38.0 to 5.8) PT: −0.55, (−22.5 to 21.4), CBT: −7.79, (−26.9 to 11.3)
	3.2.2. Workplace and patient-oriented interventions		134	Hazard Ratio (95% CI)	1.9 (1.2 to2.8)
3.3. Spinal pain	Cognitive behavioral intervention and two forms of information	1	243	Mean difference *p*-value	<0.05, RR = 9.3

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
