# Peer review of "Chronic Diseases and Employment: Which Interventions Support the Maintenance of Work and Return to Work among Workers with Chronic Illnesses? A Systematic Review"

_ijerph, 2019, doi:10.3390/ijerph16101864_

Reviewer 1 Report

I detected some language mistakes, so I suggest a proofreading performed by a mother tongue expert.

There are too many keywords… and the most important (systematic review) is not in the list!

Line 44: “over the age of 15”. Why “over 15”? Since the article focuses on the research that have considered “over 18”, it would be interesting to look at “over 18” data.

Line 45: it sounds strange that only “Ninety thousand people 45 died in 2015 due to chronic diseases worldwide”; are you sure?

Use of abbreviations is not correct through all the article. Please remember that:

- after that you introduce an abbreviation (i.e. RTW for “return to work” line 68) it’s mandatory use it and you can’t use the entire expression (see line 75)

- you don’t have to repeat the meaning of an abbreviation (i.e. lines 30 and 36; please note: this is only one example, there are many and many changes to do)

- an excessive use of abbreviations (see lines 245-255) makes the text difficult to read.

I have to be clear: you have to do many corrections about abbreviation, and it won’t be easy. You'll have to work with patience and accuracy.

Lines 153 and 155: it’s not clear what is the difference between the “Table legend” and the “Question legend”.

Line 168: it’s better to say that the Figure 1 presents “the research workflow” than “the result”.

Citations in the text are often uncorrected: you don’t have to indicate the publication year, but only the [number]. I.E.

-  line 140 (The Christensen et al. study [30]): correct

-  lines 187-188 (the study by Bergström et al., 2012 [26]): uncorrect. Please note that this mistake is often repeated through all the article.

Lines 305-312: use the ; to separate the different bullets.

Line 328: “However, the risk of long-term sick leave was only nine times lower …”. Only??? Why do you use “only”?

Line 353: Discussion are not for results, please don’t repeat results (i.e. line 381; line397; etc.).

Author Response

Rebuttal letter

ijerph-477382

Chronic diseases and employment: which interventions support the maintenance of work and return to work among workers with chronic illnesses? A systematic review

Dear reviewer,

We thank you for the critical and very helpful comments.

Response to questions of reviewer #1

I detected some language mistakes, so I suggest a proofreading performed by a mother tongue expert.

Answer: The text was proofreading by a native speaker.

There are too many keywords… and the most important (systematic review) is not in the list!

Answer: Many thanks for this note. We deleted some synonyms keywords and added systematic review.

Line 44: “over the age of 15”. Why “over 15”? Since the article focuses on the research that have considered “over 18”, it would be interesting to look at “over 18” data.

Answer: You are right, this statement is maybe a bit unclear, but the population we studied is included there. We have reworded the sentence „More than one third of the European employable population has a chronic illness and two out of three people reaching retirement age have two chronic diseases”.

Line 45: it sounds strange that only “Ninety thousand people 45 died in 2015 due to chronic diseases worldwide”; are you sure?

Answer: Many thanks for the note: the error is corrected “About15 million women and men in age between 30 and 70 died through NCDs every year worldwide”

Use of abbreviations is not correct through all the article. Please remember that:

- after that you introduce an abbreviation (i.e. RTW for “return to work” line 68) it’s mandatory use it and you can’t use the entire expression (see line 75)

Answer: Many thanks for the note. We changed it.

- you don’t have to repeat the meaning of an abbreviation (i.e. lines 30 and 36; please note: this is only one example, there are many and many changes to do)

- an excessive use of abbreviations (see lines 245-255) makes the text difficult to read.

Answer: Many thanks for the note. These abbreviations are necessary for understanding the tables S1 and S2.

I have to be clear: you have to do many corrections about abbreviation, and it won’t be easy. You'll have to work with patience and accuracy.

Answer: Thank you for this very helpful comment.

Lines 153 and 155: it’s not clear what is the difference between the “Table legend” and the “Question legend”.

Answer: Numbers from 1 to 11 in the table header are questions of the CASP checklist which itemized in the Question legend.

Line 168: it’s better to say that the Figure 1 presents “the research workflow” than “the result”.

Answer: You are right, we changed it.

Citations in the text are often uncorrected: you don’t have to indicate the publication year, but only the [number]. I.E.

- line 140 (The Christensen et al. study [30]): correct

- lines 187-188 (the study by Bergström et al., 2012 [26]): uncorrect. Please note that this mistake is often repeated through all the article.

Answer: Thank you for this very helpful comment: this mistake were changed throughout the text.

Line 328: “However, the risk of long-term sick leave was only nine times lower …”. Only??? Why do you use “only”?

Answer: Thank you for this very helpful comment. Text has been changed “The risk of long-term sick leave was reduced only for the intervention group by nine times compared to the information groups”.

Line 353: Discussion are not for results, please don’t repeat results (i.e. line 381; line397; etc.).Answer: Thank you for this very helpful comment. Text has been changed

Reviewer 2 Report

This is an important and timely review topic. The review objective is clearly stated “to identify and analyze which interventions support the maintenance of work and return to work 75 among workers with chronic illnesses”. However I have some methodological concerns that should be addressed before the manuscript is ready to publish.

Methodological concerns:

First of all the authors suggest they have used PRISMA but have not provided the completed table for review.

Search: only three databases were searched which is a minimum according to PRISMA but given the topic it is necessary to justify why additional occupational health and safety sources were not included. The search strategy as shown does not appear as comprehensive as it could be with no specific disease/disorder terms listed. It is likely that many relevant studies were missed.

Inclusion/exclusion: please define what you mean by evaluation study. Some further justification for why cohort and case control studies were excluded as they could provide useful evidence on this topic.

Selection of studies: good to see that 2 reviewers were used in all steps however excluding references when no abstract is available is not good practice. Those without abstracts should proceed to full article review as potentially relevant.

Quality assessment: Please justify why you chose to use a tool that did not provide scores and then score it. There are a host of QA tools available that do provide scores. Please be consistent in the use of low vs poor scores.

Data extraction/synthesis: Was a meta-analysis the original plan? If so please mention it in this section with a clear indication of what criteria would allow for the meta-analysis. In the discussion there is a suggestion that a meta-analysis was desired.

Additional comments:

The search yield seems very low for this topic – see comments about search strategy above. Also in the discussion the authors note that the search was extensive, I do not agree.

You have provided a breakdown of country of publication in both the text and table please choose one or the other no need to have in both. Also clarify if the Netherlands has 5 or 6 studies.

At one point in the results the authors mention not predefining secondary outcomes and yet have chosen to include them. This should be detailed in the methods with a clear indication of how they were included in the analysis. At present the reader cannot tell. In the discussion the secondary outcomes were emphasized.

Discussion: I don’t think the Netherlands is a Nordic country

Please be consistent in describing your review – is it a systematic review or is it an overview? An overview does not have the same rigor.

It is unclear to me why the authors suggest that multidisciplinary approaches are promising when there seems to be little evidence presented in the review?

The authors make a statement that including only RCTs guarantees a high methodological quality which is not strictly speaking true.

Please edit for English as there are a few places where tense and word choice make it difficult to understand what is intended.

As I mentioned above, the review topic is important and if conducted with rigor would be a useful addition to the literature. However, the manuscript requires major revisions to address the methodological concerns described above.

Author Response

Rebuttal letter

ijerph-477382

Chronic diseases and employment: which interventions support the maintenance of work and return to work among workers with chronic illnesses? A systematic review

Dear reviewer,

We thank you for the critical and very helpful comments.
Response to questions of reviewer #2
Methodological concerns:

First of all the authors suggest they have used PRISMA but have not provided the completed table for review.

Answer: Many thanks for the note. The PRISMA checklist is available on request from the authors

Search: only three databases were searched which is a minimum according to PRISMA but given the topic it is necessary to justify why additional occupational health and safety sources were not included. The search strategy as shown does not appear as comprehensive as it could be with no specific disease/disorder terms listed. It is likely that many relevant studies were missed.

Answer: We used the 3 most usual databases. Since this review was done as part of the European project Joint Action CHRODIS Plus, this methodology was a consent of all members of this project.

Inclusion/exclusion: please define what you mean by evaluation study. Some further justification for why cohort and case control studies were excluded as they could provide useful evidence on this topic.

Answer: You are right, we delated this sentence. Line: 114 “Randomized controlled trials (RCTs), controlled clinical trials (CCTs) and evaluation studies were included, and they were selected...” Then we have not included evaluation studies and we have deleted this mistake in the method. We wanted only the highest level of evidence and only studies with high internal validity. Also this topic was done as part of the European Joint Action project CHRODIS Plus and an agreement with the project partners. This fact was noted in the discussion section as a limitation of the review.

Selection of studies: good to see that 2 reviewers were used in all steps however excluding references when no abstract is available is not good practice. Those without abstracts should proceed to full article review as potentially relevant.

Answer: Thank you for the important comment. We only searched for RCTs, published RCTs, according to our experience, always have an abstract, we changed the sentence Line 123 "When the title and abstract did not fulfill one or more selection criteria or the abstract was not available, the record was excluded." in the text, since we have really not have done this.

Quality assessment: Please justify why you chose to use a tool that did not provide scores and then score it. There are a host of QA tools available that do provide scores. Please be consistent in the use of low vs poor scores.

Answer: You are right, it is a host of QA tools available, but since this review was done as part of the European Joint Action CHRODIS Plus project, it was a consent of all members of this project CHRODIS to use CASP.

Data extraction/synthesis: Was a meta-analysis the original plan? If so please mention it in this section with a clear indication of what criteria would allow for the meta-analysis. In the discussion there is a suggestion that a meta-analysis was desired.

Answer: We did not plan a meta-analysis, because we expected a heterogeneity of studies, which was also confirmed, so a meta-analysis would not have made sense.

Additional comments:

The search yield seems very low for this topic – see comments about search strategy above. Also in the discussion the authors note that the search was extensive, I do not agree.

Answer: Many thanks for the note. First: The goal of this review a priory was chronic diseases in general and RTW. Second we delate Line 439: “First, even though our search was extensive, we cannot be sure, that all relevant articles were included. Second, the choice to include only RCTs …. “

You have provided a breakdown of country of publication in both the text and table please choose one or the other no need to have in both. Also clarify if the Netherlands has 5 or 6 studies.

Answer: Many thanks for the note, the error has been corrected. Line 175: “The 15 randomized controlled studies RCTs included in this report were published between 2000 and 2017, mostly 12 in Europe (mostly in Netherlands, n=5), one in USA, one in South Africa and one in China  (Hong Kong).(s. table 2)”

At one point in the results the authors mention not predefining secondary outcomes and yet have chosen to include them. This should be detailed in the methods with a clear indication of how they were included in the analysis. At present the reader cannot tell. In the discussion the secondary outcomes were emphasized.

 Answer: Many thanks for the comment. Methodology was supplemented… Line 146: “Extracted were the study characteristics (author, year, country, study design), the patient characteristics (chronic disease diagnosis, number of included participants, age of participants, gender of participants, education and employment status, primary and secondary outcomes, follow-up) were…”

Discussion: I don’t think the Netherlands is a Nordic country

Answer: You are right, many thanks for the note. Error was corrected

Please be consistent in describing your review – is it a systematic review or is it an overview? An overview does not have the same rigor.

Answer: Many thanks for the note: The term has been changed. Line 360 summary instead of overview

It is unclear to me why the authors suggest that multidisciplinary approaches are promising when there seems to be little evidence presented in the review?

Answer: Many thanks for the note. We made the statement more understandable. Two high-quality studies (de Buck et al. 2005, Lambeek et al. 2010) and a systematic review by Sabariego et al., 2018 confirm this statement.

The authors make a statement that including only RCTs guarantees a high methodological quality which is not strictly speaking true.

RCTs have per se an internal validity. Is the methodological quality of the RCTs really good enough, so we check this with the quality assessment by CASP.

Please edit for English as there are a few places where tense and word choice make it difficult to understand what is intended.

Answer: The text was proofreading by a native speaker.

Reviewer 3 Report

Row 77 & following: a statement on progressive or degenerative chronic disease may help the understanding of whic population is concerned.

row 90 to 98: itis not clear why they belong to the objective chapter, seems to me they are irrilevant or belonging to the introduction.

Row 103 Search strategy: no mention has been made of PeDro database which comprises most of the work done on rehabilitation. A search using the same terms used for the other databases should be done to verify if any relevant paper is missing. It is surprising that no papers regarding specific rehabilitation programs with physical therapists have been found/discussed given the amount of literature in the field of muscoloskeletal disorders, a comment should be made.

Author Response

Rebuttal letter

ijerph-477382

Chronic diseases and employment: which interventions support the maintenance of work and return to work among workers with chronic illnesses? A systematic review

Dear reviewer,

We thank you for the critical and very helpful comments.

Response to questions of reviewer #3

Row 77 & following: a statement on progressive or degenerative chronic disease may help the understanding of whic population is concerned.

Answer: We target especially metabolic vascular and/or respiratory diseases as well as musculoskeletal and mental and neurology diseases. These are the most influence common diseases with great potential to the process “return to work”. This review was done as part of the European Joint Action CHRODIS Plus project, it was a consent to admit this common diseases in our review.

Row 90 to 98: itis not clear why they belong to the objective chapter, seems to me they are irrilevant or belonging to the introduction.

Answer: Yes you all right, many thanks for the comment. This information is a part of introduction. We changed it.

Row 103 Search strategy: no mention has been made of PeDro database which comprises most of the work done on rehabilitation. A search using the same terms used for the other databases should be done to verify if any relevant paper is missing. It is surprising that no papers regarding specific rehabilitation programs with physical therapists have been found/discussed given the amount of literature in the field of muscoloskeletal disorders, a comment should be made.

Answer: We used the 3 most usual databases. Since this review was done as part of the European project Joint Action CHRODIS Plus, this methodology was a consent of all members of this project.

English language and style are fine/minor spell check required

Answer: The text was proofreading by a native speaker.

Round  2

Reviewer 2 Report

The authors have addressed many of the concerns that I raised in my initial review. There are just a few minor but important issues that should be addressed to strengthen the paper.  Below are my replies to the authors’ rebuttal to the first review.  I have numbered the comments that require attention to avoid confusion.

First of all the authors suggest they have used PRISMA but have not provided the completed table for review.

Answer: Many thanks for the note. The PRISMA checklist is available on request from the authors

REPLY: 1) As the PRISMA table is usually provided for peer-review could you please provide the PRISMA table for review?

Search: only three databases were searched which is a minimum according to PRISMA but given the topic it is necessary to justify why additional occupational health and safety sources were not included. The search strategy as shown does not appear as comprehensive as it could be with no specific disease/disorder terms listed. It is likely that many relevant studies were missed.

Answer: We used the 3 most usual databases. Since this review was done as part of the European project Joint Action CHRODIS Plus, this methodology was a consent of all members of this project.

REPLY: 2) This justification/reason should be provided in the manuscript so the reader understands why the search was limited to these three databases.

Inclusion/exclusion: please define what you mean by evaluation study. Some further justification for why cohort and case control studies were excluded as they could provide useful evidence on this topic.

Answer: You are right, we delated this sentence. Line: 114 “Randomized controlled trials (RCTs), controlled clinical trials (CCTs) and evaluation studies were included, and they were selected...” Then we have not included evaluation studies and we have deleted this mistake in the method. We wanted only the highest level of evidence and only studies with high internal validity. Also this topic was done as part of the European Joint Action project CHRODIS Plus and an agreement with the project partners. This fact was noted in the discussion section as a limitation of the review.

REPLY: OK

Selection of studies: good to see that 2 reviewers were used in all steps however excluding references when no abstract is available is not good practice. Those without abstracts should proceed to full article review as potentially relevant.

Answer: Thank you for the important comment. We only searched for RCTs, published RCTs, according to our experience, always have an abstract, we changed the sentence Line 123 "When the title and abstract did not fulfill one or more selection criteria or the abstract was not available, the record was excluded." in the text, since we have really not have done this.

REPLY: 3) There are a number of reasons why abstracts may not be available for review which are not related to the study design. However if there were no instances where an abstract was missing then this is fine.

Quality assessment: Please justify why you chose to use a tool that did not provide scores and then score it. There are a host of QA tools available that do provide scores. Please be consistent in the use of low vs poor scores.

Answer: You are right, it is a host of QA tools available, but since this review was done as part of the European Joint Action CHRODIS Plus project, it was a consent of all members of this project CHRODIS to use CASP.

REPLY: 4) Please provide the reason for using the CASP in the manuscript so the reader knows why this tool was used.

Data extraction/synthesis: Was a meta-analysis the original plan? If so please mention it in this section with a clear indication of what criteria would allow for the meta-analysis. In the discussion there is a suggestion that a meta-analysis was desired.

Answer: We did not plan a meta-analysis, because we expected a heterogeneity of studies, which was also confirmed, so a meta-analysis would not have made sense.

REPLY: 5) If you did not plan to conduct a quantitative synthesis why do you say you were unable to do so in the discussion?

The search yield seems very low for this topic – see comments about search strategy above. Also in the discussion the authors note that the search was extensive, I do not agree.

Answer: Many thanks for the note. First: The goal of this review a priory was chronic diseases in general and RTW. Second we delate Line 439: “First, even though our search was extensive, we cannot be sure, that all relevant articles were included. Second, the choice to include only RCTs …. “

REPLY: OK but still seems quite low

You have provided a breakdown of country of publication in both the text and table please choose one or the other no need to have in both. Also clarify if the Netherlands has 5 or 6 studies.

Answer: Many thanks for the note, the error has been corrected. Line 175: “The 15 randomized controlled studies RCTs included in this report were published between 2000 and 2017, mostly 12 in Europe (mostly in Netherlands, n=5), one in USA, one in South Africa and one in China  (Hong Kong).(s. table 2)”

REPLY: OK

At one point in the results the authors mention not predefining secondary outcomes and yet have chosen to include them. This should be detailed in the methods with a clear indication of how they were included in the analysis. At present the reader cannot tell. In the discussion the secondary outcomes were emphasized.

 Answer: Many thanks for the comment. Methodology was supplemented… Line 146: “Extracted were the study characteristics (author, year, country, study design), the patient characteristics (chronic disease diagnosis, number of included participants, age of participants, gender of participants, education and employment status, primary and secondary outcomes, follow-up) were…”

REPLY: OK

Discussion: I don’t think the Netherlands is a Nordic country

Answer: You are right, many thanks for the note. Error was corrected

REPLY: OK

Please be consistent in describing your review – is it a systematic review or is it an overview? An overview does not have the same rigor.

Answer: Many thanks for the note: The term has been changed. Line 360 summary instead of overview

REPLY: OK

It is unclear to me why the authors suggest that multidisciplinary approaches are promising when there seems to be little evidence presented in the review?

Answer: Many thanks for the note. We made the statement more understandable. Two high-quality studies (de Buck et al. 2005, Lambeek et al. 2010) and a systematic review by Sabariego et al., 2018 confirm this statement.

REPLY: OK

The authors make a statement that including only RCTs guarantees a high methodological quality which is not strictly speaking true.

RCTs have per se an internal validity. Is the methodological quality of the RCTs really good enough, so we check this with the quality assessment by CASP.

REPLY: 6) Exactly, so your statement that RCTs guaranteed high methodological quality is untrue.  Please change the line to read “…include only RCTs guaranteed a strong study design …” Which is true.

Please edit for English as there are a few places where tense and word choice make it difficult to understand what is intended.

Answer: The text was proofreading by a native speaker.

REPLY: OK

Author Response

Reply to Reviewer 2

Many, many heartfelt thanks for your good notes and comments, they are very valuable and helpful.

The authors have addressed many of the concerns that I raised in my initial review. There are just a few minor but important issues that should be addressed to strengthen the paper.  Below are my replies to the authors’ rebuttal to the first review.  I have numbered the comments that require attention to avoid confusion.

First of all the authors suggest they have used PRISMA but have not provided the completed table for review.

Answer: Many thanks for the note. The PRISMA checklist is available on request from the authors

REPLY: 1) As the PRISMA table is usually provided for peer-review could you please provide the PRISMA table for review?

Answer: You are absolutely right, PRISMA checklist table is inserted.

Search: only three databases were searched which is a minimum according to PRISMA but given the topic it is necessary to justify why additional occupational health and safety sources were not included. The search strategy as shown does not appear as comprehensive as it could be with no specific disease/disorder terms listed. It is likely that many relevant studies were missed.

Answer: We used the 3 most usual databases. Since this review was done as part of the European project Joint Action CHRODIS Plus, this methodology was a consent of all members of this project.

REPLY: 2) This justification/reason should be provided in the manuscript so the reader understands why the search was limited to these three databases.

Answer: For the better understanding we have added the following text: Line 102 “Since this review was done as part of the CHRODIS Plus European Joint Action project, all members of this project agreed on the methodology, and decided to search the three main medical databases: PubMed, EMBASE and PsycINFO.

Inclusion/exclusion: please define what you mean by evaluation study. Some further justification for why cohort and case control studies were excluded as they could provide useful evidence on this topic.

Answer: You are right, we deleted this sentence. Line: 114 “Randomized controlled trials (RCTs), controlled clinical trials (CCTs) and evaluation studies were included, and they were selected...” Then we have not included evaluation studies and we have deleted this mistake in the method. We wanted only the highest level of evidence and only studies with high internal validity. Also this topic was done as part of the European Joint Action project CHRODIS Plus and an agreement with the project partners. This fact was noted in the discussion section as a limitation of the review.

REPLY: OK

Selection of studies: good to see that 2 reviewers were used in all steps however excluding references when no abstract is available is not good practice. Those without abstracts should proceed to full article review as potentially relevant.

Answer: Thank you for the important comment. We only searched for RCTs, published RCTs, according to our experience, always have an abstract, we changed the sentence Line 123 "When the title and abstract did not fulfill one or more selection criteria or the abstract was not available, the record was excluded." in the text, since we have really not have done this.

REPLY: 3) There are a number of reasons why abstracts may not be available for review which are not related to the study design. However if there were no instances where an abstract was missing then this is fine.

Answer: OK

Quality assessment: Please justify why you chose to use a tool that did not provide scores and then score it. There are a host of QA tools available that do provide scores. Please be consistent in the use of low vs poor scores.

Answer: You are right, it is a host of QA tools available, but since this review was done as part of the European Joint Action CHRODIS Plus project, it was a consent of all members of this project CHRODIS to use CASP.

REPLY: 4) Please provide the reason for using the CASP in the manuscript so the reader knows why this tool was used.

Answer: For the better understanding we have added the following text: Line 129. “Within the framework of the CHRODIS Plus project, two systematic reviews were conducted: 1) this review on RTW and chronic diseases in general, and 2) a previously published review on RTW and cancer [11]. The review of RTW among cancer patients did not find enough RCTs, so the colleagues decided to also include other study designs. Otherwise, both reviews should, as far as possible, use a common methodology. For this reason, it was decided that both systematic reviews should assess methodological quality using the Critical Appraisal Skills Program (CASP) 2017 checklists, because these are available for various study designs.”

Data extraction/synthesis: Was a meta-analysis the original plan? If so please mention it in this section with a clear indication of what criteria would allow for the meta-analysis. In the discussion there is a suggestion that a meta-analysis was desired.

Answer: We did not plan a meta-analysis, because we expected a heterogeneity of studies, which was also confirmed, so a meta-analysis would not have made sense.

REPLY: 5) If you did not plan to conduct a quantitative synthesis why do you say you were unable to do so in the discussion?

Answer:  You are right, the text sentence is deleted “Moreover, due to the heterogeneity of the participants in the included studies, we were unable to offer a quantitative synthesis of the outcomes.”

The search yield seems very low for this topic – see comments about search strategy above. Also in the discussion the authors note that the search was extensive, I do not agree.

Answer: Many thanks for the note. First: The goal of this review a priory was chronic diseases in general and RTW. Second we delate Line 439: “First, even though our search was extensive, we cannot be sure, that all relevant articles were included. Second, the choice to include only RCTs …. “

REPLY: OK but still seems quite low

You have provided a breakdown of country of publication in both the text and table please choose one or the other no need to have in both. Also clarify if the Netherlands has 5 or 6 studies.

Answer: Many thanks for the note, the error has been corrected. Line 175: “The 15 randomized controlled studies RCTs included in this report were published between 2000 and 2017, mostly 12 in Europe (mostly in Netherlands, n=5), one in USA, one in South Africa and one in China  (Hong Kong).(s. table 2)”

REPLY: OK

At one point in the results the authors mention not predefining secondary outcomes and yet have chosen to include them. This should be detailed in the methods with a clear indication of how they were included in the analysis. At present the reader cannot tell. In the discussion the secondary outcomes were emphasized.

 Answer: Many thanks for the comment. Methodology was supplemented… Line 146: “Extracted were the study characteristics (author, year, country, study design), the patient characteristics (chronic disease diagnosis, number of included participants, age of participants, gender of participants, education and employment status, primary and secondary outcomes, follow-up) were…”

REPLY: OK

Discussion: I don’t think the Netherlands is a Nordic country

Answer: You are right, many thanks for the note. Error was corrected

REPLY: OK

Please be consistent in describing your review – is it a systematic review or is it an overview? An overview does not have the same rigor.

Answer: Many thanks for the note: The term has been changed. Line 360 summary instead of overview

REPLY: OK

It is unclear to me why the authors suggest that multidisciplinary approaches are promising when there seems to be little evidence presented in the review?

Answer: Many thanks for the note. We made the statement more understandable. Two high-quality studies (de Buck et al. 2005, Lambeek et al. 2010) and a systematic review by Sabariego et al., 2018 confirm this statement.

REPLY: OK

The authors make a statement that including only RCTs guarantees a high methodological quality which is not strictly speaking true.

RCTs have per se an internal validity. Is the methodological quality of the RCTs really good enough, so we check this with the quality assessment by CASP.

REPLY: 6) Exactly, so your statement that RCTs guaranteed high methodological quality is untrue.  Please change the line to read “…include only RCTs guaranteed a strong study design …” Which is true.

Answer: Many thanks for the note, the text has been changed. Line 423 “Some limitations should be mentioned. Although the choice to include only RCTs guaranteed a strong study design, it did not allow us to consider many observational studies that also focused on the effectiveness of interventions, with the consequent loss of some results”.

Please edit for English as there are a few places where tense and word choice make it difficult to understand what is intended.

Answer: The text was proofreading by a native speaker.

REPLY: OK

Round  3

Reviewer 2 Report

I feel that the authors have sufficiently addressed my concerns. The manuscript is a contribution to the field.